∂ | **Open Peer Review** | Antimicrobial Chemotherapy | Observation

# Role of the two-component system AmgRS in early resistance of *Pseudomonas aeruginosa* to cinnamaldehyde

Eline Dubois,[1] Vladimir Spasovski,[1] Patrick Plésiat,[1] Catherine Llanes[1]

**ABSTRACT** Exposure of *Pseudomonas aeruginosa* to cinnamaldehyde (CNA), a natural electrophilic antimicrobial often used as self-medication to treat mild infections, triggers overproduction of the MexAB-OprM efflux system, leading to multidrug resistance. In this study, we demonstrate that CNA exposure induces expression of genes regulated by the two-component system AmgRS. AmgRS activates MexAB-OprM production, independent of repressors MexR and NalD. In addition to the essential role played by the NalC-ArmR pathway in this adaptive process, AmgRS is critical for the survival of *P. aeruginosa* challenged with CNA. Altogether, these data suggest that efflux-dependent and -independent mechanisms are activated in the early phase of CNA exposure, allowing for progressive enzymatic reduction of the biocide to non-toxic cinnamic alcohol.

**IMPORTANCE** Exposure of *Pseudomonas aeruginosa* to cinnamaldehyde (CNA), an antimicrobial used in self-medication, induces overproduction of the MexAB-OprM efflux system, leading to multidrug resistance. Our study demonstrates that the AmgRS two-component system aids in the survival of *P. aeruginosa* strain PA14 under CNA exposure through both MexAB-OprM-dependent and -independent mechanisms until the enzymatic reduction of CNA into the less toxic cinnamic alcohol. This discovery highlights the pivotal role of AmgRS in mediating defense against aldehyde biocides, emphasizing its significance in the persistence of *P. aeruginosa*, a pathogen associated with hospital-acquired infections and cystic fibrosis, and underscores the potential impact on clinical treatment strategies.

**KEYWORDS** *Pseudomonas aeruginosa*, cinnamaldehyde, two-component regulatory systems, efflux pumps, antibiotic resistance, cross resistance antibiotics essential oils

Cinnamaldehyde (CNA), the main component of cinnamon essential oil, exhibits a remarkable activity against a large variety of pathogens regardless of their susceptibility to conventional antibiotics (1). For this reason, CNA is used as self-medication to treat mild infections despite toxicity concerns that still need to be addressed more thoroughly. The strong electrophilic attack caused by the aldehyde group of this volatile molecule inflicts pleiotropic and still not fully characterized damages in bacterial cells (2). One of the main features of aldehyde-type antiseptics in general is to promote the cross-linking of proteins, RNA and DNA, and thereby an aggregation of intracellular constituents (3). At inhibitory concentrations, the cell envelope is damaged with subsequent leakage of ions and cytoplasmic molecules, and depolarization of the inner membrane (4). In contrast to many other Gram-negative species, *Pseudomonas aeruginosa* can resist quite elevated concentrations of CNA (e.g., greater than 700 µg/mL) through a complex adaptive response that primarily involves the enzymatic conversion of CNA into the less toxic cinnamic alcohol and its further degradation into uncharacterized metabolites (5). However, shortly after the initial exposure to the phytoaldehyde

Address correspondence to Catherine Llanes, catherine.llanes-barakat@univ-fcomte.fr.

The authors declare no conflict of interest.

and before its significant degradation, *P. aeruginosa* exhibits a transient burst of efflux activity (from $t_{15\ min}$ to $t_{60\ min}$ post-exposure) involving at least four multidrug efflux systems of the Resistance Nodulation, cell Division (RND) family, namely MexAB-OprM, MexCD-OprJ, MexEF-OprN, and MexXY(OprM) (5). Gene deletion experiments demonstrated that only MexAB-OprM tends to limit the CNA-promoted killing of *P. aeruginosa* in the early times of exposure, until more efficient detoxification mechanisms are initiated, such as metabolic conversion of CNA into less toxic cinnamic alcohol compound (5). This induction is concomitant with the activation of the NalC pathway in which a small protein, called ArmR, binds and sequesters MexR, the local repressor of *mexAB-oprM* (6), which eventually leads to the operon overexpression (Fig. 1). Preliminary transcriptomic experiments on CNA-treated (350 µg/mL CNA for 30 min) cells of *P. aeruginosa* wild-type reference strain PA14 highlighted overexpression of several genes controlled by the two-component system (TCS) AmgRS including a locus of unknown function annotated PA5528 in reference strain PAO1, and PA14_72930 in PA14 (Table S1) (7, 8). Since AmgRS is known to also control *mexAB-oprM* activity in cells treated with diamide, a CNA-like electrophilic stressor that among other damages causes the misfolding and aggregation of proteins (9), we searched to understand the protective role of this TCS in the survival of *P. aeruginosa* to CNA and activation of the MexAB-OprM pump.

## Exposure to CNA induces overexpression of PA5528, a gene regulated by AmgRS

Induction of the PA5528 ortholog in CNA-treated PA14 cells was checked by Reverse Transcription-quantitative PCR (RT-qPCR) experiments to confirm our transcriptomic data. Relative ratios of gene expression, normalized to gene *rpsL* whose expression is not affected by CNA treatment, were determined 30 min, 1 h, and 2 h after the addition of 350 µg/mL CNA by using primers specifically annealing to PA5528 (Table S2), as previously described (5). Consistent with our transcriptomic data, a 2.15-fold overexpression of the gene at $t_{1h}$ was recorded (Fig. 2). This ratio was close to that reported previously in gentamicin-treated PAO1 cells (10), confirming the activation of AmgRS upon CNA stress. To further investigate the role of the TCS in first-line defense of *P. aeruginosa* against the biocide, we deleted the *amgRS* operon in PA14 (PA14_68700–680). The defective mutant PA14Δ*amgRS* was constructed by using the suicide vector pKNG101 (Table S3). The DNA sequences flanking the region to be deleted were amplified with appropriate primers (Table S2) and inserted into plasmid pKNG101 by assembly cloning using the NEBuilder Hi-Fi DNA Assembly Cloning kit (New England Biolabs, France). The resultant construct was introduced by transformation into competent CC118λ*pir Escherichia coli* cells (Table S3) and then transferred in PA14 by conjugation according to a protocol reported previously (5). Excision of the undesired pKNG101 sequence was obtained by plating transformants on minimal M9 plates containing 5% (wt/vol) sucrose (14). Finally, DNA sequencing confirmed the deletion of the two-gene locus in PA14Δ*amgRS*. As expected, RT-qPCR experiments demonstrated complete loss of PA5528 induction in this mutant treated with 350 µg/mL CNA, compared with its wild-type parent (Fig. 2).

## Impact of the TCS AmgRS on the susceptibility to CNA

As assessed by a spot test on Mueller-Hinton agar (MHA) after 18 h of incubation at 37°C, mutant PA14Δ*amgRS* proved to be more susceptible to 500 µg/mL CNA than PA14 (Fig. 3A). Confirming these results, the mutant showed delayed growth in Mueller-Hinton broth (MHB) supplemented with the same concentration of CNA but recovered that of PA14, 8 to 9 h after the start of the challenge (Fig. 3B). Finally, time-kill experiments carried out on strains rendered bioluminescent by chromosomal insertion of plasmid pUC18T-MiniTn7-P1-*lux* (Table S3) (15) provided evidence that the lack of *amgRS* results in a significant decline in living cells within the first hour of exposure to 700 µg/mL CNA (maximum of $-1.6 \times Log_{10}$ RLU [relative light units] versus $-0.38 \times Log_{10}$ RLU for PA14) (Fig. 3C). As reported previously (5), inactivation of the pump MexAB-OprM (PA14Δ*mexAB*, Table S3) also sensitized PA14 to CNA, though to a greater extent ($-2.25 \times$

**A**

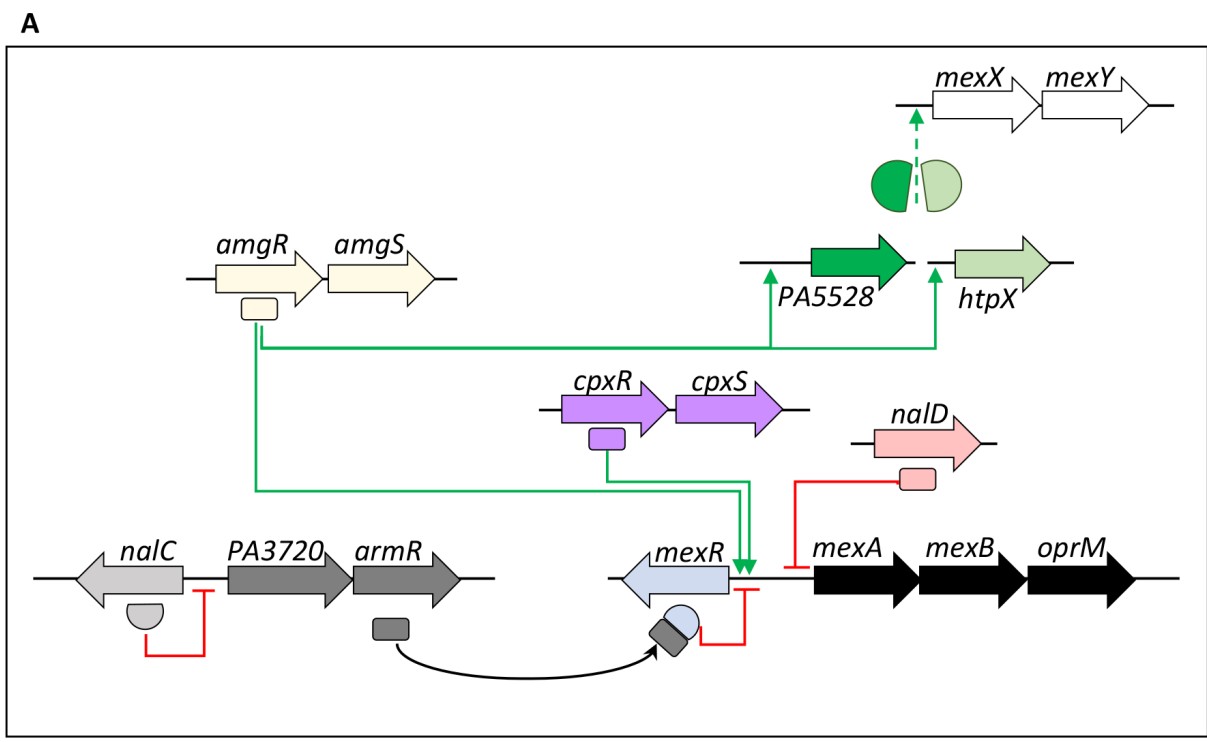

**B**

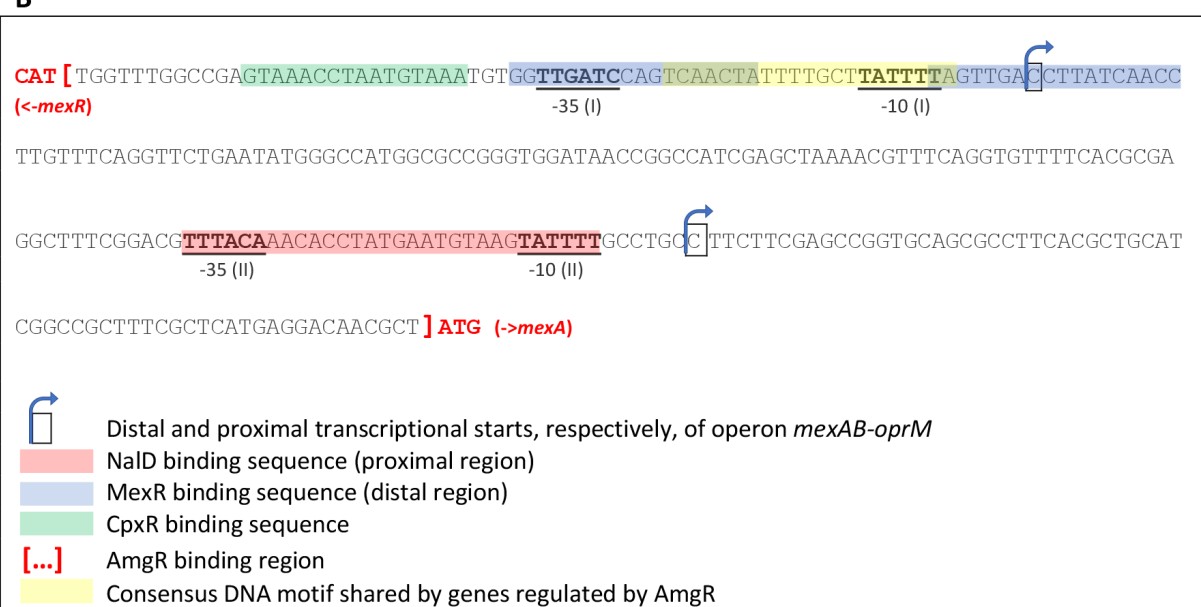

**FIG 1** Regulation of operon *mexAB-oprM* expression. (A) In the absence of inducer (e.g., CNA), repressor NalC prevents the production of the MexR antirepressor ArmR, thus allowing MexR to negatively control *mexAB-oprM* expression. In the presence of sub-Minimal Inhibitory Concentration (sub-MIC) CNA, *mexAB-oprM* is upregulated by both AmgR and ArmR/NalC (black arrow). AmgRS is also known to indirectly regulate operon *mexXY* expression through the activity of membrane protease HtpX and protein PA5528 (10). (B) Map of the *mexR-mexA* intergenic region showing the binding sites of regulators NalD, MexR, CpxR, and AmgR, respectively, as determined by electromobility shift assays (9, 11–13). The following repeated DNA motifs GNAANANNNNGNAANA and GTAAANNNNNGTAAAN have been found by *in silico* analysis upstream of the genes regulated by AmgR and CpxR, respectively (7). Alignment of the AmgR DNA motif with the region upstream of *mexAB-oprM* identified a sequence (indicated in yellow), overlapping the distal promoter of the operon and the DNA site recognized by MexR (The MEME Suite, meme-suite.org).

Log$_{10}$ RLU). As for the wild-type strain, a regrowth was observed for PA14$\Delta$*amgRS* at $t_{1h}$ due to enzymatic reduction of CNA into cinnamic alcohol and other catabolites (5).

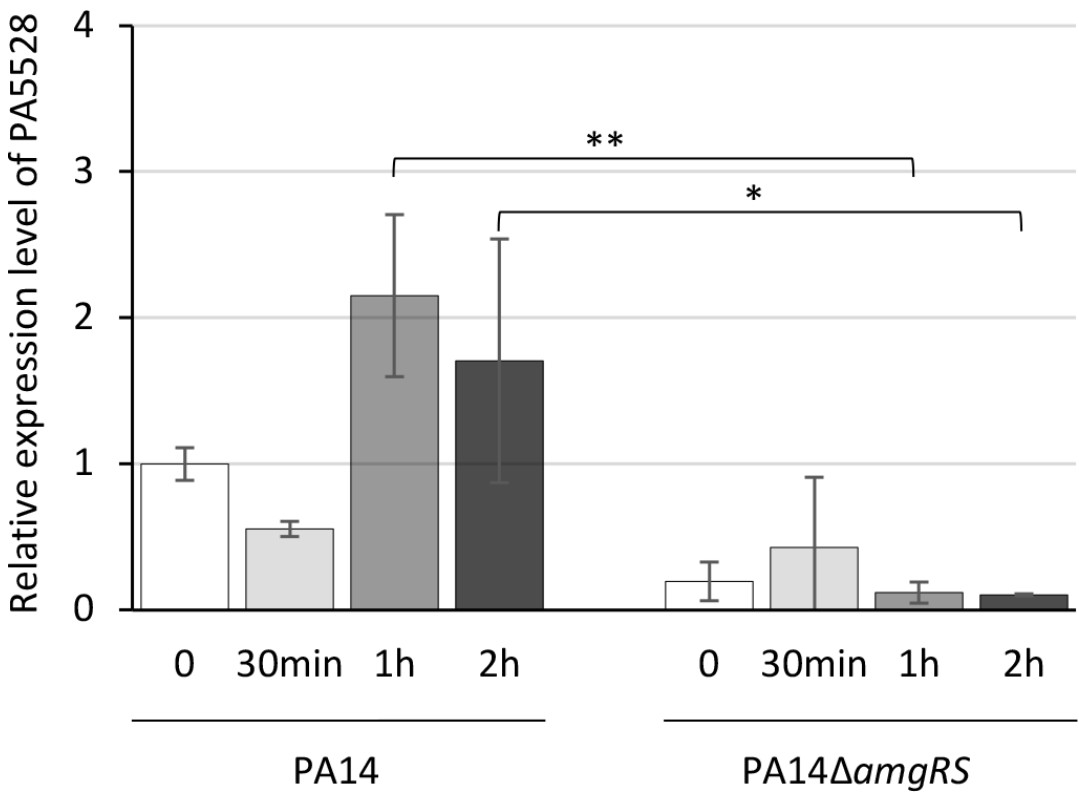

**FIG 2** Relative expression levels of gene PA5528. The relative expression levels of PA5528 in *P. aeruginosa* strains PA14 and PA14Δ*amgRS* were determined at $t_0$ (white bars), $t_{30\ min}$ (light gray), $t_{1h}$ (dark gray), and $t_{2h}$ (black) post-exposure to 350 µg/mL CNA in exponentially growing bacteria ($A_{600\ nm}$ of 0.75). The amounts of specific cDNA were assessed on a Rotor Gene RG6000 instrument (Qiagen, Courtaboeuf, France) by using the QuantiTect SYBR green PCR kit (Qiagen). The values from three biological replicates were averaged for each strain, normalized to that of the housekeeping gene *rpsL,* and finally expressed as a ratio to the transcript levels of untreated wild-type PA14 strain cultured in Mueller-Hinton broth (MHB) supplemented with 0.3% dimethylsulfoxide (DMSO), the solvent used for CNA solubilization. An analysis of variance test was performed on the data, followed by a Dunnett test comparing each time to t0 in PA14 and in PA14Δ*amgRS*, and comparing PA14Δ*amgRS* to PA14 at each selected time. A Friedman test followed by a Wilcoxon test were made for both strains at t2h. \**P*-value <0.05; \*\**P*-value <0.01.

## Overexpression of *mexAB-oprM* by CNA is partially AmgRS-dependent

As MexAB-OprM activity plays a critical role in the adaptation process of *P. aeruginosa* to CNA, we hypothesized whether CNA induces the expression of operon *mexAB-oprM* via the TCS AmgRS, like in the response to diamide exposure (9). Reminiscent of the observations made on this other electrophilic stressor, deletion of the *amgRS* genes reduced by two- to four-fold but did not totally suppress the CNA-associated induction of *mexAB-oprM* and *mexR* at $t_{30\ min}$ (Fig. 4A and B; Fig. S1A). Only the deletion of *armR*, the gene encoding the antirepressor of MexR (ArmR), abolished these activations. Previous results demonstrated that all the genes belonging to the NalC regulatory pathway known to control *mexAB-oprM* transcription (*nalC, armR, mexR*) are transiently activated in cells treated with the phytoaldehyde (5). Here, we found that the loss of *armR,* but especially *amgRS*, had a negative impact on repressor gene *nalC* expression at $t_{30\ min}$ post-exposure (Fig. S1B). Whether AmgRS downregulates *nalC* expression directly or indirectly remains to be clarified. The effects of AmgRS on *nalD* levels, another regulatory gene of *mexAB-oprM*, were non-significant (Fig. S1C). Altogether, these results pointed to an intricate activation of the operon implying AmgRS and the NalC-ArmR-MexR circuit upon CNA stress. To test this hypothesis, we therefore deleted *armR* from PA14Δ*amgRS*. The resultant double mutant PA14Δ*amgRS*Δ*armR* (PA14ΔΔ) exhibited gene expression kinetics like those of PA14Δ*armR* regarding *mexB, oprM, mexR, nalC,* and *nalD*, thereby demonstrating that AmrR is the primary regulator of *mexAB-oprM* activity in

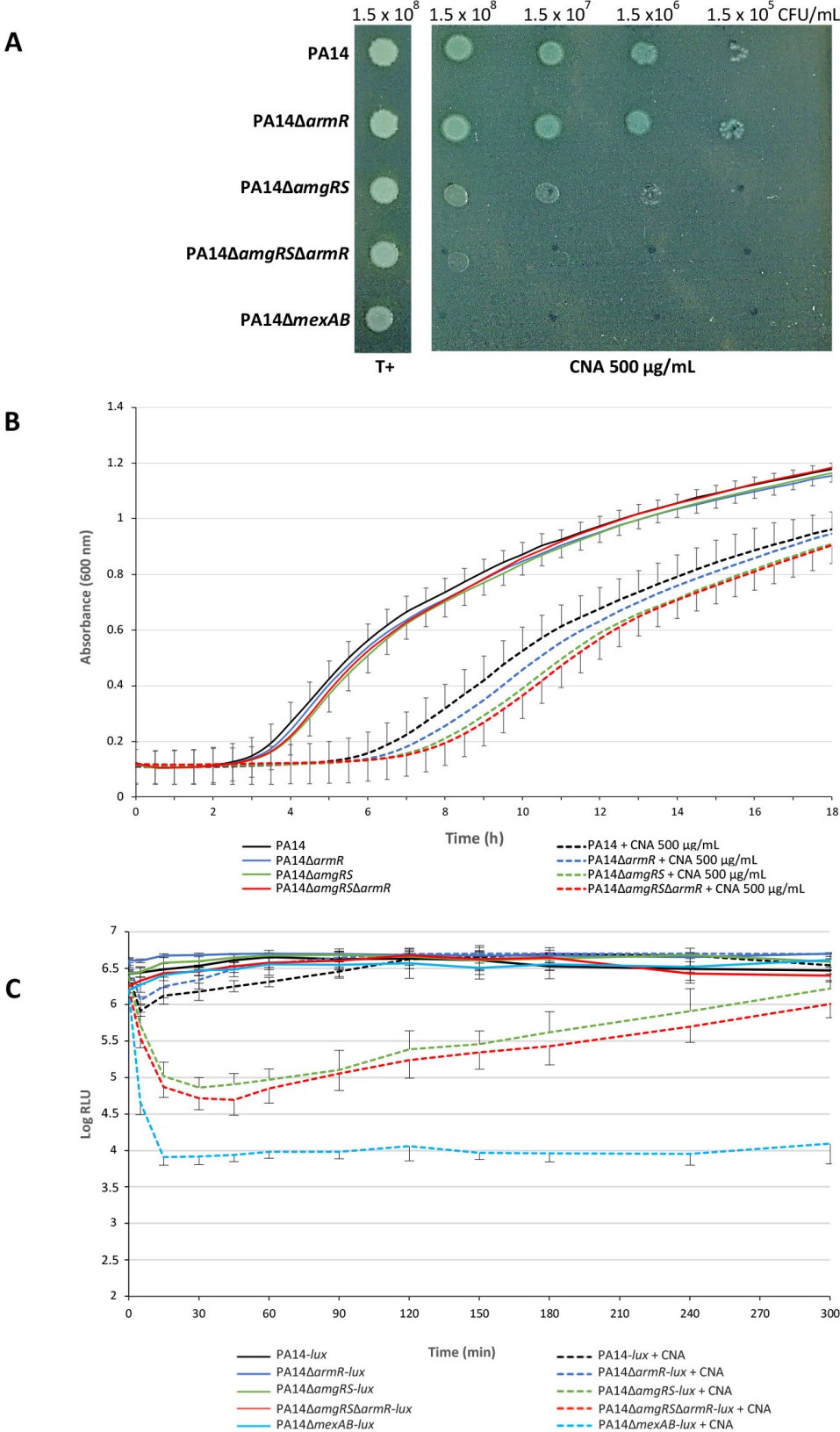

**FIG 3** Susceptibility of strain PA14 and derivative mutants to CNA. (A) Spot test where 5 µL volumes of serial dilutions of log-phase bacteria (from ca. $10^8$ to $10^5$ CFU/mL) were deposited on the surface of a Mueller-Hinton agar (MHA) plate containing 500 µg/mL CNA, and then incubated for 18 h at 37°C. Growth controls (T+) were spotted on an MHA without CNA. (Continued on next page)

**Fig 3 (Continued)**

(B) Bacterial suspensions adjusted to an $A_{600\,nm}$ of 0.01 were cultured in MHB in 96-well microplates for 18 h at 37°C, with (dashed lines) or without (solid lines) 500 µg/mL CNA. Bacterial density was assessed at $A_{600\,nm}$ in a spectrophotometer Spark (Tecan, Männedorf, Switzerland) equipped with a humidity cassette to prevent desiccation. (C) Bioluminescence (expressed in Log RLU, Relative Light Units) of bacteria developing in MHB with (dashed lines) or without (solid lines) 700 µg/mL CNA was monitored over a 4-h time course in white 96-well assay plates (Corning, NY, USA) by using a Synergy H1 microplate reader (Biotek Instrument, Winooski, USA) set at a gain value of 160, read height of 7 mm, and integration time of 2 s. The initial inoculum was adjusted to an $A_{600\,nm}$ equal to 0.8. Growth (B) and survival (C) curves are in black for PA14-*lux*, blue for PA14Δ*armR*, green for PA14Δ*amgRS*, pale blue for PA14Δ*mexAB*, and red for PA14Δ*amgRS*Δ*armR*.

CNA-exposed *P. aeruginosa* through its binding with MexR (Fig. 4A and B; Fig. S1A through C). However, despite this pivotal role, *armR* was less important than *amgRS* in the survival of strain PA14 challenged with 700 µg/mL CNA (compare the killing curves of PA14, PA14Δ*armR*, PA14Δ*amgRS*, and PA14Δ*amgRS*Δ*armR* in Fig. 3C). Taken together, these data support the notion that in response to CNA, AmgRS contributes to *mexAB-oprM* overexpression through the NalC-ArmR-MexR circuit but also controls MexAB-OprM-independent mechanisms that are essential in CNA adaptation.

## Impact of AmgRS on *P. aeruginosa* susceptibility to antibiotics

MICs (Minimal Inhibitory Concentrations) of various antibiotics exported (ticarcillin, ceftazidime, tobramycin, gentamicin, ciprofloxacin) or not (imipenem) by the efflux systems of *P. aeruginosa* (17) were determined in triplicate by the standard serial twofold microdilution method (18) in the presence of 0, 175, and 350 µg/mL of CNA, respectively, with starting inoculums adjusted to $10^5$ CFU/mL in MHB. Upon CNA challenge, resistance to the MexAB-OprM substrates ticarcillin, ceftazidime, and ciprofloxacin increased from two- to eight-fold in wild-type strain PA14, as compared with non-treated cells (Table 1). Interestingly, CNA had no effect on the drug susceptibility of mutants PA14Δ*amgRS* and PA14Δ*amgRS*Δ*armR* except a modest twofold increase in ciprofloxacin MIC, which is also a substrate for several other efflux pumps including MexEF-OprN and MexXY(OprM) (5, 19). In agreement with previous data (7), deletion of operon *amgRS* rendered PA14 four- to eightfold more susceptible to the aminoglycosides, tobramycin and gentamicin (fourfold decrease), while abolishing the antagonistic effects of CNA. Suppression of gene *armR* impaired the development of resistance to ticarcillin and ceftazidime only (≤2-fold). Collectively, these results demonstrate that AmgRS is key to activation of efflux system MexAB-OprM in *P. aeruginosa* stressed by CNA. As discussed previously (5), the low bacterial inoculums used in the above susceptibility tests were more than 1,000-fold less than in RT-qPCR experiments, resulting in a much slower degradation of CNA into cinnamic alcohol, and thus a sustained activation of AmgRS over 18 h. It should be noted that the resistance level to imipenem, a compound not actively transported by Mex pumps, was unaffected by the addition of CNA to any of the strains tested (Table 1).

## Conclusion

This work provides clear evidence that the envelope stress and/or aggregation of proteins caused in *P. aeruginosa* by the electrophilic biocide CNA activates AmgRS-dependent mechanisms of defense within the first minutes of exposure, which are further complemented by the enzymatic reduction of CNA aldehyde group. In support to this scenario, AmgRS is known to control a regulon that includes genes coding for membrane proteases (e.g., HtpX) and protease modulating factors (20). The high proteolytic activity triggered in response to AmgRS activation is critical to remove mistranslated and/or aggregated peptides from the membranes of cells impaired by ribosome inhibitors such as aminoglycosides (20, 21). Our assumption is that this increased proteolysis has the same protective function in bacteria exposed to aldehydes like diamide and CNA. In *E. coli*, chaperonins DnaK and Hsp70, which are involved in the correct folding of target proteins, have been shown to be significantly upregulated

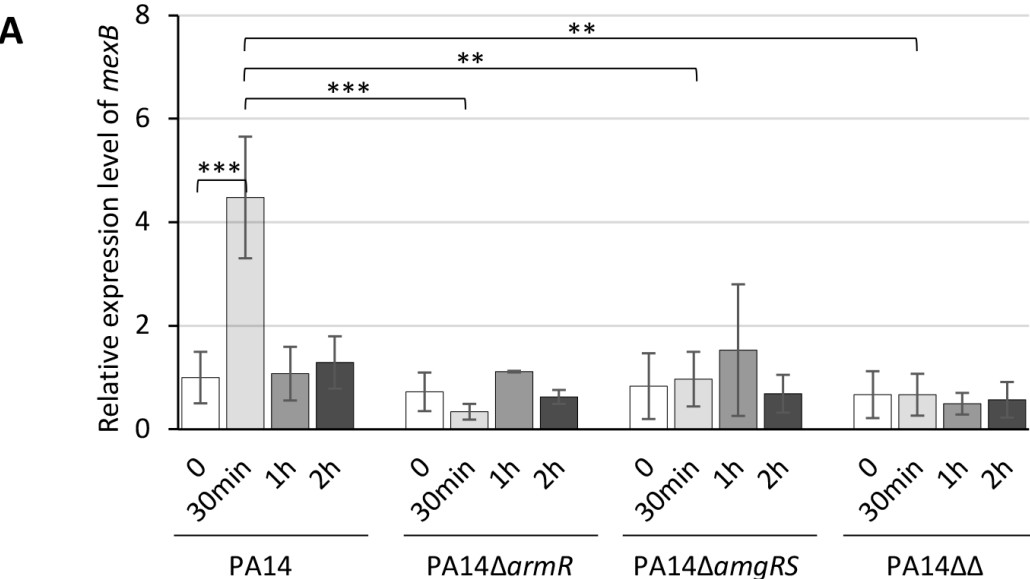

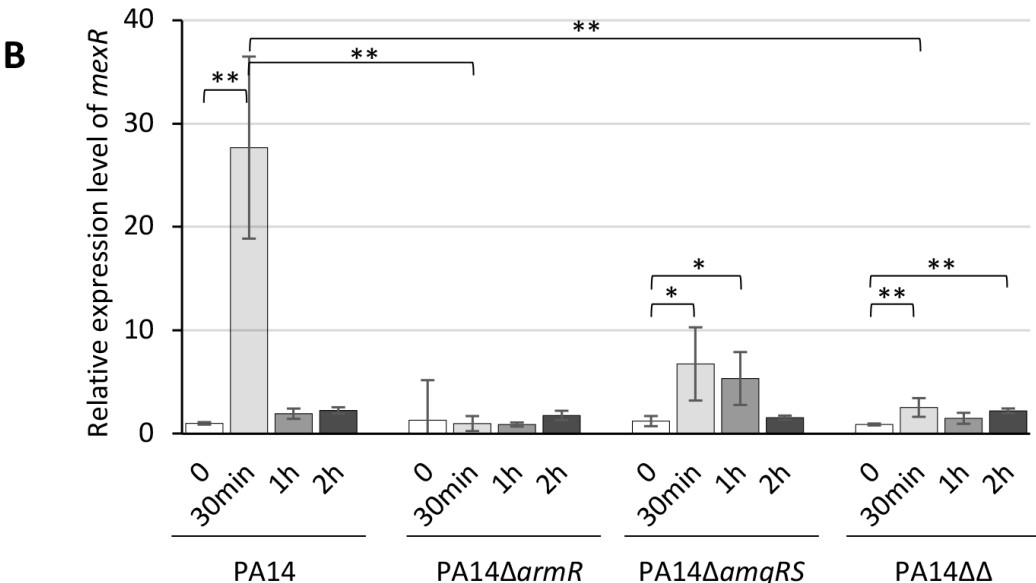

FIG 4 Relative expression levels of genes *mexB* (A) and *mexR* (B). Log-phase suspensions of strains PA14, PA14Δ*armR*, PA14Δ*amgRS*, and PA14Δ*amgRS*Δ*armR* (PA14ΔΔ) were diluted in MHB to an $A_{600\ nm}$ of 0.03 and then exposed to 350 µg/mL CNA when the absorbance reached 0.75 units. Total RNA was extracted at $t_0$ (white bars), $t_{30\ min}$ (light gray), $t_{1h}$ (dark gray), and $t_{2h}$ (black) of culture at 37°C, and mRNA levels were then quantified by RT-qPCR on a Rotor Gene RG6000 instrument (Qiagen, Courtaboeuf, France) by using the QuantiTect SYBR green PCR kit (Qiagen) on three biological replicates, averaged for each strain, normalized to that of housekeeping gene *rpsL,* and finally expressed as a ratio (Log fold change between values) to the transcript levels of untreated wild-type PA14. As established previously, strains were considered to significantly upregulate *mexB* when the gene expression level was ≥2.5-fold higher than that of PA14 (16). An analysis of variance test was performed on the data, followed by a Dunnett test comparing each time to t0, and comparing each strain to PA14 at each selected time. A Friedman test followed by a Wilcoxon test were made for PA14 and PA14Δ*armR* strain for *mexR*. *$P$-value <0.05; **$P$-value < 0.01; ***$P$_value < 0.001.

**TABLE 1** Impact of CNA on antibiotic MICs in PA14 and derivative mutants

| Strains | + CNA (µg/mL)[a] | MIC (µg/mL) of antibiotic substrates from specific efflux pumps | | | | | |
|---|---|---|---|---|---|---|---|
| | | MexAB-OprM | | MexXY(OprM) | | 4 RND[b] | No efflux[c] |
| | | Ticarcillin | Ceftazidime | Tobramycin | Gentamicin | Ciprofloxacin | Imipenem |
| PA14 | 0 | 32 | 1 | 0.25 | 0.5 | 0.125 | 1 |
| | 175 | **64** | **2** | 0.25 | 0.5 | **0.25** | 1 |
| | 350 | **128** | **4** | **0.5** | **1** | **0.5** | 1 |
| PA14ΔamgRS | 0 | 64 | 1 | 0.06 | 0.125 | 0.125 | 1 |
| | 175 | 64 | 1 | 0.06 | 0.125 | **0.25** | 1 |
| | 350 | 64 | 1 | 0.06 | 0.125 | **0.25** | 1 |
| PA14ΔarmR | 0 | 32 | 1 | 0.25 | 0.5 | 0.125 | 1 |
| | 175 | 32 | 1 | 0.25 | 0.5 | **0.25** | 1 |
| | 350 | 32 | **2** | **0.5** | **1** | **0.5** | 1 |
| PA14ΔamgRSΔarmR | 0 | 32 | 1 | 0.06 | 0.06 | 0.125 | 1 |
| | 175 | 32 | 1 | 0.06 | 0.06 | **0.25** | 1 |
| | 350 | 16 | 1 | 0.03 | 0.06 | **0.25** | 1 |

[a]CNA concentrations, 175 or 350 µg/mL, are 1/4 or 1/2 MIC, respectively; a negative control without CNA (0) containing 0.3% dimethylsulfoxide (DMSO) was made in parallel.
[b]Ciprofloxacin is the substrate for four RND pumps: MexAB-OprM, MexXY(OprM), MexCD-OprJ and MexEF-OprN, all of them being induced by CNA (5).
[c]Imipenem is not a substrate of efflux systems. **In bold**, increased resistance level due to CNA exposure (experiment done in triplicate).

in response to CNA exposure (22). On the other hand, our study shows that until CNA is converted into non-toxic degradation products, AmgRS contributes to the upregulation of efflux system MexAB-OprM in connection with the NalC-AmrR-MexR regulatory pathway. Loss of pump activity in mutant PA14Δ*mexAB* strongly sensitized strain PA14 to the biocide, supporting the notion that this relatively hydrophobic molecule is exported by the efflux system. While MexXY(OprM) plays a secondary role in the survival of *P. aeruginosa* to CNA (5), our results show that its overproduction is only marginally influenced by AmgRS and implies one or several other regulatory pathways (Fig. S2). However, as stated before, membrane proteases upregulated in response to AmgRS activation are essential in the defense of the bacterium against aminoglycosides (10). From all these observations, it appears that the two-component system AmgRS is essential to the adaptation of *P. aeruginosa* to CNA, and possibly other aldehyde biocides, through the activation of both MexAB-OprM-dependent and -independent mechanisms.

## ACKNOWLEDGMENTS

We are grateful to Lison Schmidlin for technical support and Coralie Bertheau-Rossel for statistical analysis.

This work was supported by a grant from the Region "Bourgogne Franche-Comté" and the "Centre de Recherche Pseudomonas".

## AUTHOR AFFILIATION

[1]UMR CNRS 6249 Chrono-Environnement, UFR Santé, Université Bourgogne Franche-Comté, Besançon, France

## AUTHOR ORCIDs

Patrick Plésiat https://orcid.org/0000-0001-7381-5671
Catherine Llanes http://orcid.org/0000-0001-9671-9084

## AUTHOR CONTRIBUTIONS

Eline Dubois, Conceptualization, Formal analysis, Investigation, Methodology, Writing – original draft | Vladimir Spasovski, Investigation, Methodology | Patrick Plésiat, Conceptualization, Supervision, Validation, Writing – review and editing | Catherine Llanes, Conceptualization, Funding acquisition, Investigation, Supervision, Validation, Writing – original draft, Writing – review and editing

## ADDITIONAL FILES

The following material is available online.

### Supplemental Material

**Supplemental material (Spectrum01699-24-s0001.docx).** Figures S1 and S2; Tables S1 to S3.

### Open Peer Review

**PEER REVIEW HISTORY (review-history.pdf).** An accounting of the reviewer comments and feedback.

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
