## [Reviewer comments · Microbiology Spectrum]

Microbiology Spectrum

Role of the Two-Component System AmgRS in Early Resistance of *Pseudomonas aeruginosa* to Cinnamaldehyde

Eline Dubois, Vladimir Spasovski, Patrick Plésiat, and Catherine Llanes

Corresponding Author(s): Catherine Llanes, UMR CNRS 6249 Chrono-Environnement

Review Timeline:

Submission Date:	July 24, 2024
Editorial Decision:	September 2, 2024
Revision Received:	October 9, 2024
Editorial Decision:	November 6, 2024
Revision Received:	November 7, 2024
Accepted:	November 9, 2024

Editor: Eric Cascales

Reviewer(s): The reviewers have opted to remain anonymous.

Transaction Report:

DOI: <https://doi.org/10.1128/spectrum.01699-24>

Re: Spectrum01699-24 (Role of the Two-Component System AmgRS in Early Resistance of *Pseudomonas aeruginosa* to Cinnamaldehyde)

Dear Catherine,

Thank you for submitting your work to Microbiology Spectrum. The manuscript has been sent to two external referees with expertise in the field. As you will see in their comments below, both referees recommend publication of your work, pending a few grammatical, typographical and structural modifications. They were more concerned by the lack of statistical analyses. I recommend to carefully address the points raised by the referees and invite you to submit a revision of your work.

Revision Guidelines

Sincerely,
Eric

Eric Cascales
Editor
Microbiology Spectrum

Reviewer #1 (Comments for the Author):

The work carried out by Dubois et al. explores the role of the two-component system AmgRS in the resistance of *Pseudomonas*

aeruginosa to cinnamaldehyde (CNA). Although the study relies on preliminary transcriptomic data that are not detailed, these data suggest a positive regulation of genes controlled by AmgRS in the presence of CNA. This hypothesis is partially confirmed by the observation of PA5528 gene overexpression by RT-qPCR in the presence of CNA. The authors show that Δ amgRS strains exhibit increased sensitivity to CNA and a reduction in the overexpression of mexAB-oprM induced by CNA stress, suggesting a potential role of AmgRS in the regulation of this efflux pump. They also demonstrate that the deletion of armR (anti-repressor) does not have a significant impact on the bacterium's sensitivity to CNA, although a complete loss of mexAB-oprM overexpression induced by CNA stress is observed. Finally, the authors show that mutant strains exhibit increased sensitivity to antibiotics that are substrates of the MexAB-OprM pump and conclude that the two-component system AmgRS is essential for *P. aeruginosa* adaptation to CNA, through the activation of both MexAB-OprM-dependent and -independent mechanisms.

- A more complete description and statistical analysis of the data are necessary to strengthen the conclusions.
- Figure S1A seems important for understanding the system and would merit inclusion in the main text rather than as supplementary material.
- Although the submission format is flexible, the structure chosen by the authors is unbalanced, with methods sometimes overly detailed. I would recommend a succinct description of the materials and methods in the text, with more detailed information in the figure and table legends, which would allow more space in the text for a more thorough data analysis.
- The overexpression of PA5528 in response to CNA in the wild-type strain does not necessarily indicate the activation of AmgRS under CNA stress (line 72). The complete lack of PA5528 expression in the Δ amgRS strain shows that this gene is part of the AmgRS regulon, but couldn't other regulatory factors be involved in the response to CNA?
- (line 63) The reference to Table S1 seems unnecessary.
- (line 70) Note that the times mentioned in the text do not correspond to those in Figure 1, (2-fold and 2.7-fold overexpression of PA5528 are observed after 1 hour and 2 hours, not at 30 minutes and 1 hour.)

The results obtained by Dubois et al. are promising and provide important insights into the role of the AmgRS system in the resistance of *Pseudomonas aeruginosa* to CNA. These results open up new perspectives for understanding the molecular mechanisms of CNA resistance.

Reviewer #2 (Comments for the Author):

Overall, this is a well-written manuscript that furthers our understanding of the role of both the AmgRS TCS and MexAB-OprM efflux system in the survival of *P. aeruginosa* in the presence of antimicrobial agents. Below you will find several comment/suggestions.

1. The Abstract is only two sentences and is rather short (<100 words). Consider providing a little more detail.
2. Line 32: The word "of" is missing between "regardless" and "their"
3. Lines 45 and 46, double-check spacing between the subscripts 15 min and 60 min post exposure.
4. Missing citation at the end of the sentence starting at line 44 and ending in line 47.
5. Line 49: what are the "more efficient detoxification systems" please provide an example.
6. AmgRS is thought to respond to membrane damage/stress. Is there any evidence that CNA causes membrane damage in *P. aeruginosa* and/or other bacteria? Htpx is thought to be a protease. Is there any evidence that CNA causes mis-translation and protein misfolding?
7. Line 67-69, sentence is unclear. What is meant by "amounts of specific cDNA". Do you mean relative gene expression level of PA5528? For clarity, indicate that rpsL gene was used to normalize expression levels in the main body of the text.
8. Are rpsL levels unaffected by CNA? Is there any evidence of this?
9. Line 70-72, please clarify what is meant by "ratios"?
10. Line 74: it should read "...by using the suicide vector"
11. Line 77, the word "thanks" is colloquial. Reword.
12. Line 79: by "moved" do you mean "mobilized"
13. Line 79-80: After mobilization of the deletion vector and before PCR confirmation of the deletion, please indicate how cells were cured of the vector backbone.
14. Line 80: Please clarify what is meant by "PCR-sequencing".
15. Line 80-82, there are two separate ideas in this sentence. Consider revising.
16. Line 83 Title: Be more specific and indicate that the TCS is AmgRS
17. Line 91: Define RLU at first mention.
18. Line 93: There is no mention of PA14 Δ mexAB in the TableS1 provided.
19. Line 97: Recommend using "hypothesized" rather than "wondered".
20. It appears that nalD expression is modestly inducible at 30 min and 2h and this induction is lost in the armR and amgRS single and double knockouts. Are ArmR and AmgRS known to regulate nalD expression? How is this to be interpreted?
21. If AmgRS acts via ArmR to induce mexABM gene expression, is there a putative AmgR binding site within the nalC-armR intergenic region? Can a consensus sequence in AmgR-regulated promoters be identified?
22. Statistical analyses for all gene expression results is missing. Please address.
23. Line 155: the word connexion should be "connection"

24. Line 156: "Suppression" of pump activity in the efflux pump knockout would be considered "inactivation" or "loss of pump activity". Consider rewording.

Figures/Tables/Legends

1. Legend of Figure 1: Were "mRNA amounts measured"? Based on the methods, it would appear that relative gene expression levels were determined. The word "amount" suggests absolute quantification.
2. Figure 3 Legend: Is it total RNA that was extracted or mRNA specifically?
3. Figure 2B and 2C, make sure gene names are italicised.
4. Figure S1: ArmR abrogates binding of MexR to the mexABM promoter, but this is not clear in the Figure. Currently, it appears that ArmR must interact with MexR to repress expression of mexABM. Consider revising.
5. Figure S1. The relationship between AmgR and HtpX and PA5528 is not clear in the figure. Has AmgR been shown to directly regulate the genes encoding for HtpX and PA5528? Consider revising to show that AmgR governs expression of HtpX and PA5528. Also, consider using different colors for the PA5528 and HtpX proteins.
6. Table 1 Caption, Line 225, "substate" should be "substrate".
7. Table 1: Consider indicating the efflux pump substrates in the Table itself rather than in the caption. This would improve readability.

For all:

The work carried out by Dubois et al. explores the role of the two-component system AmgRS in the resistance of *Pseudomonas aeruginosa* to cinnamaldehyde (CNA). Although the study relies on preliminary transcriptomic data that are not detailed, these data suggest a positive regulation of genes controlled by AmgRS in the presence of CNA. This hypothesis is partially confirmed by the observation of PA5528 gene overexpression by RT-qPCR in the presence of CNA. The authors show that Δ amgRS strains exhibit increased sensitivity to CNA and a reduction in the overexpression of mexAB-oprM induced by CNA stress, suggesting a potential role of AmgRS in the regulation of this efflux pump. They also demonstrate that the deletion of armR (anti-repressor) does not have a significant impact on the bacterium's sensitivity to CNA, although a complete loss of mexAB-oprM overexpression induced by CNA stress is observed. Finally, the authors show that mutant strains exhibit increased sensitivity to antibiotics that are substrates of the MexAB-OprM pump and conclude that the two-component system AmgRS is essential for *P. aeruginosa* adaptation to CNA, through the activation of both MexAB-OprM-dependent and -independent mechanisms.

- A more complete description and statistical analysis of the data are necessary to strengthen the conclusions.
- Figure S1A seems important for understanding the system and would merit inclusion in the main text rather than as supplementary material.
- Although the submission format is flexible, the structure chosen by the authors is unbalanced, with methods sometimes overly detailed. I would recommend a succinct description of the materials and methods in the text, with more detailed information in the figure and table legends, which would allow more space in the text for a more thorough data analysis.
- The overexpression of PA5528 in response to CNA in the wild-type strain does not necessarily indicate the activation of AmgRS under CNA stress (line 72). The complete lack of PA5528 expression in the Δ amgRS strain shows that this gene is part of the AmgRS regulon, but couldn't other regulatory factors be involved in the response to CNA?
- (line 63) The reference to Table S1 seems unnecessary.
- (line 70) Note that the times mentioned in the text do not correspond to those in Figure 1, (2-fold and 2.7-fold overexpression of PA5528 are observed after 1 hour and 2 hours, not at 30 minutes and 1 hour.)

The results obtained by Dubois et al. are promising and provide important insights into the role of the AmgRS system in the resistance of *Pseudomonas aeruginosa* to CNA. These results open up new perspectives for understanding the molecular mechanisms of CNA resistance.

For the Editor:

I recommend accepting this article with revisions.

The authors should perform statistical analyses to strengthen their conclusions.

Although the article is well-written, improving the structure (balance between materials and methods and results) would also significantly enhance the value of this study.

Reviewer #1 (Comments for the Author)

The work carried out by Dubois et al. explores the role of the two-component system AmgRS in the resistance of *Pseudomonas aeruginosa* to cinnamaldehyde (CNA). Although the study relies on preliminary transcriptomic data that are not detailed, these data suggest a positive regulation of genes controlled by AmgRS in the presence of CNA. This hypothesis is partially confirmed by the observation of PA5528 gene overexpression by RT-qPCR in the presence of CNA. The authors show that Δ amgRS strains exhibit increased sensitivity to CNA and a reduction in the overexpression of mexAB-oprM induced by CNA stress, suggesting a potential role of AmgRS in the regulation of this efflux pump. They also demonstrate that the deletion of armR (anti-repressor) does not have a significant impact on the bacterium's sensitivity to CNA, although a complete loss of mexAB-oprM overexpression induced by CNA stress is observed. Finally, the authors show that mutant strains exhibit increased sensitivity to antibiotics that are substrates of the MexAB-OprM pump and conclude that the two-component system AmgRS is essential for *P. aeruginosa* adaptation to CNA, through the activation of both MexAB-OprM-dependent and -independent mechanisms.

1. A more complete description and statistical analysis of the data are necessary to strengthen the conclusions.

Thank you for this request. The statistical analysis was performed on Δ Ct values as described by Ganger et al., (2017). An Anova was used when data normality and heteroscedasticity were confirmed; otherwise, a Friedman test was applied. These were followed by a Dunnett's test after the Anova, and a Wilcoxon test after the Friedman test, when the p_value was significant ($p < 0.05$). Comparisons were made between time t0 and other time points within the same strain for the same gene using paired samples. A second comparison was made at one time point between PA14 and other strains for the same gene. Information on the statistical analysis has been added to the legends of figures concerned.

2. Figure S1A seems important for understanding the system and would merit inclusion in the main text rather than as supplementary material.

Done, Figure S1 is now included in the main text as Figure 1.

3. Although the submission format is flexible, the structure chosen by the authors is unbalanced, with methods sometimes overly detailed. I would recommend a succinct description of the materials and methods in the text, with more detailed information in the figure and table legends, which would allow more space in the text for a more thorough data analysis.

Thank you for this suggestion. Materials and methods have been added to the figure legends, except for the construction of inactivated mutants, which pertains to all figures and has been left in the main text.

4. The overexpression of PA5528 in response to CNA in the wild-type strain does not necessarily indicate the activation of AmgRS under CNA stress (line 72). The complete lack of PA5528 expression in the Δ amgRS strain shows that this gene is part of the AmgRS regulon, but couldn't other regulatory factors be involved in the response to CNA?

Among the eleven genes known to be regulated by AmgRS (*htpX*, *yegH*, *sugE*, *ygiT*, *yccA*, *yceJ*, *yebE*, *nlpD*, PA5528, *mexAB*, *mexXY*), *htpX*, PA5528, *yccA*, *yebE*, *mexAB* and *mexXY* are overexpressed in response to CNA exposure. This strongly suggests the involvement of this two-component system. A table (Table S1) was added to the supplementary data to show the relative expression levels of genes regulated by AmgRS after exposure to CNA (from our transcriptomic data).

Previous analyses have showed that other regulatory factors such as NalC and CmrA are implicated in efflux pump activation (MexAB-OprM and MexEF-OprN, respectively). However, to our knowledge, no other regulators have been reported to control PA5528.

5. (line 63) The reference to Table S1 seems unnecessary.

Done.

6. (line 70) Note that the times mentioned in the text do not correspond to those in Figure 1, (2-fold and 2.7-fold overexpression of PA5528 are observed after 1 hour and 2 hours, not at 30 minutes and 1 hour.)

Done

The results obtained by Dubois et al. are promising and provide important insights into the role of the AmgRS system in the resistance of *Pseudomonas aeruginosa* to CNA. These results open up new perspectives for understanding the molecular mechanisms of CNA resistance.

Reviewer #2 (Comments for the Author)

Overall, this is a well-written manuscript that furthers our understanding of the role of both the AmgRS TCS and MexAB-OprM efflux system in the survival of *P. aeruginosa* in the presence of antimicrobial agents. Below you will find several comment/suggestions.

1. The Abstract is only two sentences and is rather short (<100 words). Consider providing a little more detail.

Thank you for this suggestion. The abstract has been expanded and is now more detailed (120 words).

2. Line 32: The word "of" is missing between "regardless" and "their"

Done

3. Lines 45 and 46, double-check spacing between the subscripts 15 min and 60 min post exposure.

Done

4. Missing citation at the end of the sentence starting at line 44 and ending in line 47.

Done

5. Line 49: what are the "more efficient detoxification systems" please provide an example.

The idea was clarified at the end of the sentence: "Gene deletion experiments demonstrated that only MexAB-OprM tends to limit the CNA-promoted killing of *P. aeruginosa* in the early times of exposure, until more efficient detoxification mechanisms are initiated, such as metabolic conversion of CNA to the less toxic cinnamic alcohol compound (Tetard *et al.*, 2019)."

6. AmgRS is thought to respond to membrane damage/stress. Is there any evidence that CNA causes membrane damage in *P. aeruginosa* and/or other bacteria? Htpx is thought to be a protease. Is there any evidence that CNA causes mis-translation and protein misfolding?

Thank you for this comment. Membrane damage caused by CNA has been extensively documented in the literature, particularly in *E. coli* and *S. aureus*, where it increases cell permeability (Shen *et al.*, 2015, *Food Control*) and induces membrane disruption and oxidative damage (Wang *et al.*, 2019, *Arch. of Microb.*). Similar effects have also been observed in *Aeromonas hydrophila* (Pei *et al.*, 2023, *Aqua. Res.*). In *P. aeruginosa*, CNA has been shown to depolarize membranes (Topa *et al.*, 2018, *Microbiology*) and cause membrane damage leading to ultrastructural changes (Bouhdid *et al.*, 2010, *J. of Appl. Microbiol.*). Numerous studies have also reported membrane damage in fungi and plants pathogens. The reference (Bouhdid *et al.*, 2010) is in the paper.

Regarding the impact of CNA on protein synthesis, there are some papers suggesting that cinnamaldehyde induces mistranslation or protein misfolding. This compound has been shown to downregulate elongation factor G thereby affecting ribosome translocation in *Salmonella Typhimurium* (Fiori Silva *et al.*, 2018, *Res. In Microbiol.*). In *E. coli*, molecular chaperones DnaK and Hsp70, which are involved in the correct folding of target proteins are

significantly upregulated in response to CNA exposure (Lin *et al.*, 2017, *Food and Agricult. Immun.*). We added in the conclusion: In *E. coli*, chaperonins DnaK and Hsp70, which are involved in the correct folding of target proteins, have been shown to be significantly upregulated in response to CNA exposure (Lin *et al.*, 2017).

7. Line 67-69, sentence is unclear. What is meant by "amounts of specific cDNA". Do you mean relative gene expression level of PA5528? For clarity, indicate that *rpsL* gene was used to normalize expression levels in the main body of the text.

"Gene *rpsL*, whose expression is not affected by CNA treatment, was used to normalize expression levels" was added in the text.

8. Are *rpsL* levels unaffected by CNA? Is there any evidence of this?

RT-qPCR experiments are usually carried out in *P. aeruginosa* with housekeeping genes *rpsL* or *uvrD* (Dumas *et al.*, 2006, *FEMS Microbiol. Lett.*). For this study, *rpsL* was retained because it shows stable expression in PA14 and in inactivated mutants, with or without CNA. See previous question.

9. Line 70-22, please clarify what is meant by "ratios"?

Sorry for the lack of precision. The ratio corresponds to the Log fold change between values (added in the text). It is calculated as $R = 2^{-\Delta\Delta Ct}$ with $\Delta\Delta Ct = \Delta Ct_{gene\ of\ interest-rpsL}^{sample} - \Delta Ct_{gene\ of\ interest-rpsL}^{reference}$

10. Line 74: it should read "...by using the suicide vector"

Done

11. Line 77, the word "thanks" is colloquial. Reword.

Thanks was replaced by "using".

12. Line 79: by "moved" do you mean "mobilized"

"moved to" was replaced by "transferred in".

13. Line 79-80: After mobilization of the deletion vector and before PCR confirmation of the deletion, please indicate how cells were cured of the vector backbone.

Excision of pKNG101 from the chromosome of transconjugants was obtained by positive selection on M9-sucrose medium, which is lethal for cells that still harbor the *sacB* gene present on pKNG101. We added a sentence in the text and a reference: "Excision of the undesired pKNG101 sequence was obtained by plating transformants on minimal M9 medium containing 5% (wt/vol) sucrose (Kaniga *et al.*, 1991)".

14. Line 80: Please clarify what is meant by "PCR-sequencing".

The step of verification has been re-formulated and "PCR-sequencing" was replaced by "DNA sequencing".

15. Line 80-82, there are two separate ideas in this sentence. Consider revising.

Thank you for the comment: the sentence has been split into 2.

16. Line 83 Title: Be more specific and indicate that the TCS is AmgRS

Done: Impact of the TCS AmgRS on the susceptibility to CNA.

17. Line 91: Define RLU at first mention.

Done: Relative Light Units.

18. Line 93: There is no mention of PA14 Δ mexAB in the TableS1 provided.

We apologise for this oversight: strains PA14 Δ mexAB and PA14 Δ mexAB-lux, have been added in Table S3 (re-numbered) with their corresponding reference.

19. Line 97: Recommend using "hypothesized" rather than "wondered".

Done

20. It appears that nalD expression is modestly inducible at 30 min and 2h and this induction is lost in the armR and amgRS single and double knockouts. Are ArmR and AmgRS known to regulate nalD expression? How is this to be interpreted?

Thanks for the comment. Indeed, Figure S1C shows that *nalD* expression is lower when *amgRS* or/and *armR* are inactivated. However, in one case (Δ *armR*) induction is only delayed, and in the other case (Δ *amgRS*), the presumed loss of induction is not statistically significant (Figure S1C). In fact, NaID has a minor impact on the bacterial response to CNA, it is only induced twice and previous studies in our laboratory have shown that PA14 Δ *nalD* exhibited the same susceptibility as PA14 to CNA. The NaID binding region has been precisely defined on the *mexAB-oprM* proximal promoter (which is different from that of MexR and that of AmgR, Figure 1), and there is no evidence for a regulation of *nalD* by ArmR or AmgR. Induction of *nalD* appears to be independent of AmgR.

We wrote in the text: "The effects of AmgRS on *nalD*, another regulatory gene of *mexAB-oprM* were non-significant (Fig. S1C)".

21. If AmgRS acts via ArmR to induce mexABM gene expression, is there a putative AmgR binding site within the nalC-armR intergenic region? Can a consensus sequence in AmgR-regulated promoters be identified?

The binding site of AmgR between *mexR* and *mexA* is not exactly known. Electromobility shift assays (Fruci *et al.*, 2018) showed AmgR binds to the upstream region of *mexAB-oprM* (indicated into red brackets in Figure 1). Repeated DNA motif GNAANANNNGNAANA have been found by *in silico* analysis upstream of the genes regulated by AmgR (Lee *et al.*, 2009) (cited in the legend of Figure 1). A further *in silico* analysis, using The MEME Suite, allowed us to identify this motif in the distal promoter of *mexAB-oprM*. This was added in the legend of the Figure 1: "Alignment of the AmgR DNA motif with the upstream region of *mexAB-oprM* identified a sequence (shown in yellow) overlapping the distal promoter of the operon and the DNA site recognized by MexR (The MEME Suite, meme-suite.org)."

22. Statistical analyses for all gene expression results is missing. Please address.

All the relative gene expression figures (Fig. 2 PA5528; Fig. 4A and 4B *mexB* and *mexR*; Fig. S1A, Fig. S1B and Fig. S1C *oprM*, *nalC* and *nalD*, and Fig. S2 *mexY*) were redone and supplemented by statistical analysis (described in the legends).

23. Line 155: the word connexion should be "connection"

Done

24. Line 156: "Suppression" of pump activity in the efflux pump knockout would be considered "inactivation" or "loss of pump activity". Consider rewording.

Done

Figures/Tables/Legends

1. Legend of Figure 1: Were "mRNA amounts measured"? Based on the methods, it would appear that relative gene expression levels were determined. The word "amount" suggests absolute quantification.

Thank you for the comment: "The mRNA amounts of PA5528" was changed and replaced by "Relative expression levels of PA5528 [...]".

2. Figure 3 Legend: Is it total RNA that was extracted or mRNA specifically?

We have corrected: "Total RNA was extracted at t_0 (white bars), $t_{30 \text{ min}}$ (light grey), $t_{1 \text{ h}}$ (dark grey) and $t_{2 \text{ h}}$ (black) of culture at 37°C, and mRNA level was then quantified by RT-qPCR [...]."

3. Figure 2B and 2C, make sure gene names are italicised.

Done

4. Figure S1: ArmR abrogates binding of MexR to the *mexABM* promoter, but this is not clear in the Figure. Currently, it appears that ArmR must interact with MexR to repress expression of *mexABM*. Consider revising.

We fully agree with this comment. It is now accepted that ArmR "sequesters" MexR. Figure 1 has been made clearer.

5. Figure S1. The relationship between AmgR and HtpX and PA5528 is not clear in the figure. Has AmgR been shown to directly regulate the genes encoding for HtpX and PA5528? Consider revising to show that AmgR governs expression of HtpX and PA5528. Also, consider using different colors for the PA5528 and HtpX proteins.

Thank you for your comment. Our transcriptomic analysis revealed that *htpX* and PA5528 are two of the eleven genes regulated by the AmgRS system (*htpX*, *yegH*, *sugE*, *ygiT*, *yccA*, *yceJ*, *yebE*, *nlpD*, PA5528, *mexAB-oprM*, *mexXY*). AmgR has been shown to bind upstream of some of those genes (e.g., *mexAB-oprM*), with DNA consensus sequences found upstream all of these genes (Lee et al., 2009). Based on these findings, we can hypothesize that AmgR directly regulates *htpX* and PA5528. Figure 1 has been revised accordingly, and colour scheme has been updated.

6. Table 1 Caption, Line 225, "substate" should be "substrate".

Done

7. Table 1: Consider indicating the efflux pump substrates in the Table itself rather than in the caption. This would improve readability.

Done

Re: Spectrum01699-24R1 (Role of the Two-Component System AmgRS in Early Resistance of *Pseudomonas aeruginosa* to Cinnamaldehyde)

Dear Catherine,

Thank you for submitting your revised manuscript. It has been sent to the two original reviewers, who acknowledge that you properly and carefully addressed their comments. Reviewer #2 is however concerned by a specific point regarding growth curve analysis (see below), and I think this could be quickly addressed. Before I proceed to final acceptance of your manuscript, I will kindly ask you to answer this query and to submit the revised version.

Revision Guidelines

Sincerely,
Eric

Eric Cascales
Editor
Microbiology Spectrum

Reviewer #1 (Comments for the Author):

Thank you for addressing my comments. The changes regarding the statistical analysis, the adjustments to the figures, and the

revisions to the text have clarified the manuscript. These modifications help to better appreciate the significance of the results.

Reviewer #2 (Comments for the Author):

Dear authors,

Thank you for sufficiently addressing my previous comments. Please note that in the results section that the comment on "doubling time"(see line 103) is inaccurate, as no growth curve analysis was performed in the study. Please remove the comment on doubling times. Alternatively, perform growth curve analyses to determine doubling times and conduct statistical analyses. Arguably, for Figure 3B, the slopes of the exponential phase of CNA treated cells appear to be less steep than the untreated cells (i.e., doubling times may indeed be different). Thus, without proper curve analyses, it is inaccurate to comment on doubling times.

Reviewer #1 (Comments for the Author)

Thank you for addressing my comments. The changes regarding the statistical analysis, the adjustments to the figures, and the revisions to the text have clarified the manuscript. These modifications help to better appreciate the significance of the results.

Reviewer #2 (Comments for the Author)

Thank you for sufficiently addressing my previous comments. Please note that in the results section that the comment on "doubling time"(see line 103) is inaccurate, as no growth curve analysis was performed in the study. Please remove the comment on doubling times. Alternatively, perform growth curve analyses to determine doubling times and conduct statistical analyses. Arguably, for Figure 3B, the slopes of the exponential phase of CNA treated cells appear to be less steep than the untreated cells (i.e., doubling times may indeed be different). Thus, without proper curve analyses, it is inaccurate to comment on doubling times.

Thank you for your comment. We have calculated doubling times and performed statistical analysis (see below). It appears that CNA significantly affects the doubling time of all mutants, but without a significant difference between PA14 and the mutants. As a result, we have removed the comment on the doubling time as suggested: "Confirming these results, the mutant showed delayed growth in Mueller-Hinton broth (MHB) supplemented with the same concentration of CNA but recovered that of PA14, 8 to 9 h after the start of the challenge (Fig. 3B)."

Re: Spectrum01699-24R2 (Role of the Two-Component System AmgRS in Early Resistance of *Pseudomonas aeruginosa* to Cinnamaldehyde)

Dear Dr. Llanes, dear Catherine

Thank you for addressing the concern of reviewer #2. I am please to accept your manuscript for publication in Microbiology Spectrum, and I am forwarding it to the ASM production staff. Your paper will first be checked to make sure all elements meet the technical requirements. ASM staff will contact you if anything needs to be revised before copyediting and production can begin. Otherwise, you will be notified when your proofs are ready to be viewed.

Sincerely,
Eric

Eric Cascales
Editor
Microbiology Spectrum